# Determinant Factors of M&As in Emerging Economies: The Impact of Financial Performance in Romanian Minority Acquisitions

Liviu-George Maha [1], George-Marian Aevoae [1,2], Elena-Daniela Viorică [1] and Roxana-Manuela Dicu [1,*]

1   Faculty of Economics and Business Administration, Alexandru Ioan Cuza University of Iași,
    700505 Iași, Romania; mlg@uaic.ro (L.-G.M.); aevoae@gmail.com (G.-M.A.); dana.viorica@gmail.com (E.-D.V.)
2   Academy of Romanian Scientists, Ilfov 3, 050044 Bucharest, Romania
*   Correspondence: roxana.dicu@uaic.ro

**Abstract:** The paper aims at describing two dimensions of acquirers' behaviour when purchasing minority shares in Romanian listed target companies, based on a sample of 710 Romanian minority acquisitions. The first dimension regards the acquirer's decision to invest a certain amount, being influenced by the profitability of the target company The relationship was found to be positive and significant. To test the model further, a sample of 308 transactions was used, after excluding the transactions involving primary sector and blue-chip target companies. The second dimension focuses on the amount of purchased stake, which leads to either financial gains or the takeover intention, under the influence of the target company's operational profit. The results show a positive and significant relationship for the small stakes and a non-significant one for the high stakes.

**Keywords:** minority acquisitions; dividends; Bucharest Stock Exchange; financial performance; emerging economies

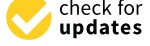



## 1. Introduction

Researchers around the world have taken an interest in examining the external growth strategies of companies, focusing on mergers and acquisitions (M&As), with emphasis on those involving companies located in developed economies (Park 2019; Yang and Deng 2017; Lin et al. 2009; Cheng and Yang 2017; Caiazza et al. 2017). There are also studies that prove the choice for these types of economies that host the acquiring or the target companies involved (Lucas 1990; Aevoae et al. 2019). On the other hand, globalisation and rapid economic growth determined a new strategic approach for companies, which have started to search for investment opportunities in emerging economies. As a result, the volume and number of M&As involving companies located in emerging markets recorded a notable increase, becoming comparable with developed economies (Zhou et al. 2016).

As external growth strategies, M&As are usually discussed as a whole, without a difference between mergers and acquisitions being made, due to the fact that the perspectives from which they are studied impose a common approach (when they are seen from a macroeconomic or geographic perspective) or separate (when they are analysed from a managerial or accounting point of view). Mergers and acquisitions (M&As) is an umbrella term for many types of transactions in which one organisation or the assets of one organisation become a part of another organisation or two organisations become one. On the other hand, Valdone et al. (2020) distinguish between three types of deals related to M&As: acquisitions, which involve an acquirer that purchases at least 50% plus one shares in the target company; minority stakes, representing stakes that are purchased but lower than 50%; and development capital, where the specific type of minority stake investment typically involves venture capitals or private equity firms investing in early stage companies, providing new financial resources and managerial expertise with the aim

of supporting their growth. When focusing on the acquisitions which do not lead to the control of the target companies, the motives that determine the acquirers to purchase stakes are related to sharing the profitability of the acquired company, sharing technology, or developing joint products (Nain and Wang 2018). Being used with the purpose of improving operating efficiency, the minority acquisitions may lead to reduced costs, to mitigation of the financial constraints, to dividends, to an increase in profitability, or to facilitation of innovative activities (Lee et al. 2006). Kabbach de Castro et al. (2021) discuss the impact of financial constraints in the increase in the number and value of minority acquisitions, especially in emerging economies, due to economic uncertainties and less developed capital markets. Other authors (Kengelbach et al. 2020) consider that minority deals are becoming more common, being driven by the emergence of large corporate ecosystems, in case of which the suspicion for takeover is limited or unlikely. As support for this opinion, it must be mentioned that, in 2020, this type of transaction represented 35% of all mergers and acquisitions (M&A), up from 20% in the 1990s (Macoris et al. 2023).

In our study, we intended to identify and to present a profile of the investors who purchase less than the majority stake in target companies resident in emerging economies. This research highlights the influence of a target company's performance on an acquirer's decision to purchase a minority stake with the purpose of obtaining financial gains or with the purpose of taking it over, considering the social, institutional, and economic environment found in these types of economies.

In line with these objectives, we developed our theoretical and methodological arguments in the context of emerging economies. Hoskisson et al. (2000) identified, in their study, a list of 64 emergent economies, based on criteria related to GDP, GNP per capita, and inflation rate. Khanna and Palepu (2010) draw a line between emerging economies such as BRICS, which have had an economic growth higher than the one of the developed economies, and the emerging economies as a whole, characterised by cheap workforce and resources, but also by corruption, bureaucracy, the risk of receivables non-collection, poor infrastructure, and misevaluation of investment opportunities. As officials of the International Monetary Fund (henceforth FMI), Duttagupta and Pazarbasioglu (2021) assert that emerging economies are generally identified based on such attributes as sustained market access, progress in reaching middle-income levels, and greater global economic relevance. Nenu et al. (2018) and Haroon and Rizvi (2020) consider the dimension of a stock market as a determinant for emerging economies, described by indicators such as trades, volume, market capitalisation, and number of transactions. In this respect, emerging economies are mainly characterised by a rapid pace of economic development and by governments which have adopted market-based policies that favour economic liberalisation and the adoption of a free-market system.

As a result of more than 15 years history of being a functional market economy, Romania is considered by many authors (Albu et al. 2013; Albu and Albu 2012; Borlea et al. 2017; Poenaru 2021), as well as by financial institutions (International Monetary Fund 2019), as an emerging economy. Romania is a country that passed multiple stages to reach, in September 2020, the status of a secondary emerging market, granted by FTSE Russell (FTSE Russell 2020; Dicu et al. 2019), despite its controversial evolution, proven by the delay and the various methods of privatisation applied by the Romanian government, which ultimately boosted the market for corporate control as nowhere else in Central and Eastern Europe (Pop 2006).

GDP is one of the pillar stones which made it possible for Romania to achieve the emerging economy status (World Bank 2022). The 2000s were influenced by its membership in NATO in 2004, as well as its accession to the European Union in 2007, with both events leading to major increases in GDP per capita (11.06% and 11.14%, respectively). After a robust cumulative growth in GDP of 30.82% in 2000–2005, the effects of the global economic crisis also began to manifest in Romania, resulting in GDP declining in 2009 (5.52%) and again in 2010 (3.90%). After that, in the last years, the GDP of Romania recorded a constant growth, with the maximum being reached in 2017 (7.32%) (World Bank 2022). The year

2020 was a difficult year for Romania, with the pandemic leading to a decrease of 3.86% in GDP and of 3.43% in GDP per capita. Despite these data, Romania continues to have one the lowest minimum wages in the European Union (EUR 515 gross salary in 2022), being followed only by Bulgaria (Eurostat 2022).

The second pillar, the stock market, also played an important role that allowed Romania to increase its rank towards achieving the emerging economy status, mainly because this market is representative of the FTSE Russell criteria. The Bucharest Stock Exchange (henceforth, BSE) reopened on 21 April 1995. The first years (from 1995 until 2004) were characterised by slow growth, followed in 2004 by an increase of 316.76% in volume and of 280.20% in market capitalisation. Although the GDP of Romania recorded a significant decrease in 2009, the financial market reacted in 2008, anticipating the crisis with a decrease in market capitalisation of 46.84%. After the years of the financial crisis, in 2013, the market capitalisation recorded an excellent revival of 36.95%, and it has continued to constantly grow since then. In 2022, it reached a number of 83 listed companies on the main market; 278 listed on the alternative market AeRO (Alternative Exchange in Romania); and 14 companies listed on the SMT International, dedicated to financial instruments admitted to trading on a regulated market or an equivalent market with a regulated market in a third country (Bucharest Stock Exchange 2022). Due to its role in financing economic activities, we consider, in our research, the companies listed on the BSE as target companies in acquisitions of minority interests.

This paper aims at analysing two dimensions of the behaviour of the investors on the BSE, considering twofold objectives. On one side, we have the investments made with the main purpose of generating dividends or to be sold when their market price improves. This will be considered, in this paper, portfolio investments. Given their potential of improving the acquirers' cash-flow, we wanted to assess whether the financial performance of the target company significantly influences the investment decision of the acquirer in terms of the amount spent. On the other side, we analysed whether the decision to acquire stakes in the target companies, listed on the BSE, with the intention to take over the target company, is influenced by the efficiency of the operational activity reported by the acquired company. We used a database of 710 Romanian minority acquisitions to explore the two analysed dimensions of the acquirers' behaviour.

The paper is structured in three parts. The first part reviews the scientific literature regarding the minority acquisitions, the motives that lead the acquirers to be involved in these types of transactions, and the factors that may influence the acquirer in paying a deal value for a minority stake. Also, this part includes the hypotheses to be tested and validated, regarding the determinant factors for an acquirer to pursue an investment in a target company, with the purpose of obtaining dividends or to take over the target company through multi-stage acquisitions. The second part concerns the research methodology and design. The third part of the paper focuses on the empirical results of our study.

This study contributes to minority acquisition's research (Bostan and Spătăreanu 2018; Contractor et al. 2014; Lee et al. 2006; Liao 2014; Ouimet 2013) by examining the acquirers' behaviour under the influence of performance-related factors.

## 2. Literature Review and Hypotheses

In Garzella and Fiorentino's (2017) opinion, when companies are involved in M&As, they are looking to gain synergy as an increase in future economic benefits. Thus, the performance of the target company before, during, and after the transaction is closed is a significant factor in assessing the synergistic success of M&As (Rozen-Bakher 2017; Rozen-Bakher 2018).

Beyond this objective, minority acquisitions, which refer to acquisitions of equity stakes where acquirers purchase less than 50 percent of target companies' shares (Pinelli et al. 2020; Bostan and Spătăreanu 2018; Contractor et al. 2014; Ouimet 2013), have the purpose of pursuing value creation opportunities in order to grant a certain degree of influence over the target company's decisions (Stepanov 2019) or to access assets or innovation (Gao et al.

2019; Lee et al. 2006). Thus, the concept of minority acquisitions is subject to discussion, given the fact that, in practice, there can be particularities, mostly related to the volume of shares purchased by acquirers in target companies, which may allow a specific extent of the influence (or lack thereof) of the acquiring company. Also, Contractor et al. (2014) consider minority acquisitions preferable in the cases of lower institutional distance between the nations of the involved companies and in the case of high cultural distance. In other words, when the institutional environment in the two countries is similar, but the local culture is not, the acquirers do not want to control target companies.

The reasons for an acquirer to purchase a minority stake are multiple and quite different, ranging from solving financing constrains to issues related to the access of the target companies to the capital market. According to Liao (2014), there are three motives that lead to purchasing minority stakes in target companies, motives that can be related to both acquirers and target companies: contracting, financing, and governance motives. The contracting motive is related to the relationship between supplier and customer and the way it can be improved or cover some shortcomings, mitigating incomplete contracts and facilitating cooperation between two independent companies (Lee et al. 2006). In the case of a financially constrained target company (financing motive), the results are mixed. There is evidence that the minority acquisitions work like a guarantee for the capital market or for financial institutions (Hertzel and Smith 1993), while there is research which proves that non-controlling acquisitions do not seem to alleviate these financial constraints (Urzúa 2012). Kang and Kim (2008) consider corporate blockholders (owners of stock of shares) as monitors of large shareholders, with whom they share control, a fact that gives them a governance motive to purchase minority acquisitions. Ouimet (2013) considers that minority acquisitions are more likely when the target's valuation is especially uncertain; integrating internal capital markets will be costly; and consolidating earnings will lower earnings per share.

In search of purchasing shares in a target company, an acquirer has to answer several questions: In what type of companies does it intend to invest? Is it interested in the profit or loss of the target? Should it belong to a specific sector? The deal value that is paid in an M&A depends on both target and acquirer, but the latter is the one who pays the price, and the acquired company should be the one that provides enough motives to be chosen. In this context, the target company accounting figures may justify the opportunity of an investment in terms of profitability and revenues that reflect the M&A success (Rozen-Bakher 2017; Sirower and Lipin 2003). Seeing the synergy success in terms of revenue increase is more of a managerial approach than an accounting one, given the fact that the decrease in costs may also lead to profitability in terms of accounting statements, but the revenue increasing may be connected to increase in market share (Bauer and Matzler 2013) or in innovation (Wubben et al. 2016; Aevoae et al. 2019). Also, the success of an M&A can be assessed through the degree in which the financial targets are met in a time frame, established during the pre-acquisition phase, in terms of expected returns and costs (Dilshad 2012).

Also, the acquirer has to decide if these investments are strategic or tactical (Payne 1987), depending on the outcome it expects: to take over the target company or to gain financial revenues, under the form of dividends (facts that support our two hypotheses). Considered by Urzúa (2012) non-controlling blocks, the first type of acquisitions allows for the acquirers to purchase a significant stake in the target company, which allow, in the long run, the takeover of the target company through repetitive acquisitions. Between 1995 and 2015, almost 20% of the world M&As were two-stage acquisitions, in which the acquirer first purchased a minority stake, followed by a majority stake (Vansteenkiste 2020). The main purposes of this decision were related to board representation and to the reduction of the information asymmetry between the involved parties.

In this context, we must bring into attention the fact that, in these types of transactions, the shareholders of the target company are protected by the mandatory bid rule, which allows them to benefit equally from the premium that the acquirer is willing to pay

(Psaroudakis 2010). At the European Union level, the main regulation in this regard is the 13th Directive on Company Law, which applies from 2004, on the principle *de lege lata* and *de lege fecunda*, being subject to evolution, considering the profound restructuring on the European market and the increase in the number of mergers and acquisitions on the continent (European Union 2004). In this regard, the number of deals in the last 30 years increased by 352.79% and their value increased by 364.16%, with differences between Western and Eastern Europe (i.e., in 2019, the number of reported deals in Western Europe was 15.051, while in Eastern Europe it was only 2.854 deals, which is representative for the degree of evolution between the two regions) (Institute of Mergers, Acquisitions and Alliances 2022). In the case of Romania, disclosure of material shareholding applies where a shareholder acquires or disposes of shares of an issuer listed on a regulated market and to which voting rights are attached, if the percentage of the voting rights held following the acquisition or the disposal concerned, reaches, exceeds, or falls below one of the 5%, 10%, 15%, 20%, 25%, or 33%, as a result of implementing the Transparency Directive 2004/109/EC. Also, any offer by which the bidder wishes to acquire more than 33% of the voting rights in the target company represents a voluntary takeover offer (Bondoc et al. 2023).

The situation of dividends after a merger or an acquisition is a subject of debate because it can be analysed from two perspectives: the one of the acquirer and the one of the target, given the fact that one of the parties has to collect and the other one is in the position of the payer. Thus, their attitude regarding dividends is different, and there are separate factors that can influence the decision to purchase shares in order to collect revenues versus the decision to pay dividends (Dereeper and Turki 2012).

In both cases, the performance must be brought to the front. In the case of takeover intentions, we consider the operational profit (earnings before interest and taxation) because it estimates how efficiently a company can earn profit from its assets, regardless of its size and without being affected by management financial decisions. A high value of this indicator can provide a sign of solid operational performance (Polemis and Gounopoulos 2012; Purba and Septian 2019). In the case of obtaining financial gains, the performance expressed through profit and loss is relevant as an overall result of the activity in an accounting period (Glendening et al. 2016). We controlled for size of the target company, accounting practice, and the year of the transaction.

In this context, this research was proposed to test and validate the following hypotheses:

**Hypothesis 1.** *Financial performance represents a determinant factor when acquirers are willing to invest a certain amount, as deal value, with the purpose of obtaining financial gains.*

**Hypothesis 2.** *The efficiency of operational activity is a determinant factor for an acquirer when pursuing to invest for a specific stake in a target company, with the purpose of taking it over.*

The proposed hypotheses were tested and validated using the statistical software programmes R and Eviews.

### 3. Methodology

*3.1. Target Population and Analysed Sample*

To test and to validate the two proposed research hypotheses, this study analysed the empirical data related to 710 M&As, for the 2010–2018 period, considering the acquisitions of minority interests in Romanian listed companies. The chosen period is significant for Romania. The year 2010 was when GDP and GDP per capita started to grow, after the 2008–2009 financial crisis (World Bank 2022). The year 2018 marked a historical moment for the European Union, because on the 14th of November, the Brexit Withdrawal Agreement was published, endorsed on 25 of November by 27 EU member states. The act, covering matters such as money, citizens' rights, border arrangements, and dispute resolutions, had a great impact on the economy of the EU, including Romania (Schimmelfennig 2018; Jensen and Snaith 2018). As a result of Brexit, financial services and financial technology

moved from London to other financial centres in the EU (Donnelly 2023), which led to a destabilisation of financial markets (Van Kerckhoven 2021). Thus, a decline in the number and value of M&As was observed in 2019, compared to previous years (Kengelbach et al. 2020; Lin and Chen 2020).

For the first objective, this paper explored the perspective of purchasing a tactical investment with the main purpose of collecting dividends. This dimension was analysed for both the whole sample of transactions and for a selected sample from which the transactions consisting of target companies from the primary sector—due to the fact that acquisitions are motivated by other interests than financial performance, consistent with the research of Andreff (2016)—and target companies considered to be blue-chips—in which acquisitions are only made due to the possibility of gaining dividends (these companies are considered to be national interest companies that are active in utilities and the financial sector) (Chigrinskaya 2019)—were excluded. After the aforementioned criterion was taken into consideration, our selected sample consisted of 308 target companies.

For the second objective, we analysed the intention to take over the target company. For this purpose, we split the sample of 710 transactions into two sub-samples, clustering the two categories of transactions: below and above 1% of the shares purchased by the acquirers in the target company. The threshold of 1% is significant because, in 2010, it was the limit imposed for acquisitions in listed financial investment companies to prevent their hostile takeover.

### 3.2. Variable Description and Proposed Models

We estimated two regression models, according to the papers' objectives.

Before presenting the regression models, we considered that the decision to invest in minority acquisitions (as amount to be paid or as purchased stock of shares) is a function of financial performance of the target companies, as follows:

$$Y = f \text{ (financial performance)} + \varepsilon, \tag{1}$$

The first model seeks to estimate the marginal effect of financial performance reported by the target company for the year prior to the acquisition over the deal value by employing an ordinary least square (OLS) regression.

As such, the regression equation has the following form:

$$Y = \beta_0 + \beta_1 X + \delta_i Z_i + \varepsilon, \tag{2}$$

where $Y$ is the logarithmic deal value, $X$ is the profit or loss reported by the target company, $Z_i$ is the vector of control variables described below, and $\varepsilon$ is the error term.

For the second objective, we explored the relationship between the purchased stake in minority acquisitions and the efficiency of the operational activity, reflected by earnings before interest and taxes (henceforth, EBIT), as stated by Polemis and Gounopoulos (2012) and Purba and Septian (2019). Since the distribution of purchased stake is bimodal, clearly showing different distributions for the high values (between 1 and 50%) and for the low values of stake (up to 1%), we used a quantile regression (QR) approach, specifically the conditional quantile regression estimator developed by Koenker and Hallock (2001). Compared to the OLS method of estimation, quantile regression analyses the different responses of the dependent variable to changes in the independent variables for any conditional percentile of the dependent variable. Quantile regression does not assume normality or homoscedasticity, and since, for our sample, the distribution of the dependent variable is not normal, applying conditional mean estimators to our equation would not be suitable since these estimators are not robust to departures from normality, and therefore OLS is likely to produce inefficient and biased estimates, whereas quantile regression produces more robust estimates for non-normal distributions (Mata and Machado 1996). More so, estimating a quantile regression model can prove to be an efficient technique in

better understanding the relationship between two variables, especially when dealing with outliers.

We specified the following conditional quantile regression model:

$$Y = X'\alpha\theta + \gamma\theta_iV_i + u\theta \tag{3}$$

$$\text{Quant}\theta(Y \mid X) = X'\alpha\theta + \gamma\theta_iV_i, \tag{4}$$

where Y is the value of stake, X is the operating profit or loss (EBIT), $\text{Quant}\theta(Y \mid X)$ is the $\theta$th conditional quantile of Y on the regressor X, the vector of parameters $\alpha\theta$ and $\gamma\theta$ are estimated for different values of $\theta$ in (0, 1), $\theta$ is the error term, and $V_i$ is the vector of control variables.

The variables are presented in Table 1.

**Table 1.** The variables used in the models.

| Name | Way of Calculation | Description | Sources |
|---|---|---|---|
| Deal value | Th $ | The price paid by the acquirers for the stake purchased | Zephyr database |
| Stake | % of shares purchased in the target company | Number of purchased shares, reported to total shares of the target company | Zephyr database |
| Profit and loss (P/L) | Th $ | The net result of the target companies, reported for the year prior the acquisitions | Orbis database |
| EBIT | Th $ | The efficiency of a company's capacity to earn profit from its assets | Orbis database |
| Profile of the acquirer | Dummy variable: 1. listed 0. unlisted | The position of the acquirer regarding the capital market and the requested level of transparency | Zephyr database |
| Size of the company | Ln (total assets) | The value of the assets from the financial statements | Orbis database |
| Year of the transaction | Numeric | 2010–2018 | Zephyr database |
| Accounting practice | Dummy variable: 1. IFRS 0. Local GAAP | The accounting regulations which are applied by the target companies in preparing financial statements | Orbis database |

Source: Own processing. Note: $ is US dollar.

3.2.1. Dependent Variables

We measured the investment decision of the acquirer, in terms of the amount spent, by the deal value. Deal value represents the price paid by the acquirers for the stake purchased in the target companies, and it is influenced by a series of factors that mainly determine its size (Dicu et al. 2019). Alexandridis et al. (2010) found a positive correlation between the number of acquirers and the value paid in transaction, while Alexandridis

et al. (2013) found a negative correlation between the size of the target and the deal value paid in transaction. Given the fact that the main outcomes of a M&A are synergy success and/or efficiency gains, we wanted to assess whether the profitability of the target company influences the amount the acquirer intends to invest in purchasing shares of the target company.

We used the purchased stake as a measure of intention to take over the target company. Purchased stake represents the number of shares purchased in the target companies, reported to its total number of shares, issued and outstanding (0.001–50%).

### 3.2.2. Independent Variables

The financial performance of a company is universally accepted as being reflected in the profit and loss (P/L) reported in its financial statements (Polemis and Gounopoulos 2012; Capon et al. 1990). It represents the net result of the target companies, reported for the year prior the acquisitions. The importance of pre-M&A profitability is underlined by Cantwell and Santangelo (2002), who support the influence of net income or loss on the decision of the acquirer of purchasing a specific target.

The efficiency of a company's capacity to earn profit from its assets, regardless of its size and without being affected by management financial decisions, is reflected in earnings before interest and taxes (EBIT). A high value of this indicator can provide a sign of solid operational performance (Purba and Septian 2019; Polemis and Gounopoulos 2012). Also, Gudmundsson et al. (2017) consider EBIT the measure for profitable companies involved in M&As in the pre-M&A phase.

### 3.2.3. Control Variables

We used different control variables for each hypothesis. For the first one, we used the acquirer's profile, the year of the transaction, and the purchased stake. For the second one, the control variables are the size of the target company, accounting practice, and the year of the transaction.

The profile of the acquirer refers to the three types of acquirers: listed and unlisted, with the latter also including undisclosed acquirers (usually persons who want to invest with the sole purpose of earning dividends/capital gains). This control variable is important because the acquirers have different rules regarding transparency and the capacity of participating in transactions. Undisclosed acquisitions were determined based on the fact that the acquirer was not presented in Zephyr database, because they/it decided to remain unknown.

Size of the target company is a numeric variable, calculated using a natural logarithm from the total assets of the target company for the year prior the acquisition, as accepted in the literature (Owen and Yawson 2010; Chuang 2017; Ferrouhi 2014). Since this variable can influence both of the interest variables, using it as a control provides a better understanding of the relationship between the two variables.

Year of the transaction is between 2010 and 2018 and corresponds to the year when the minority acquisition was completed. We used this control variable since each year has its own specificities from an economic, political, or legislative point of view.

Accounting practice of the target company provides an in-depth view on the market the target company is listed on: primary (IFRS) or AeRO (local GAAP).

## 4. Results and Discussions

The descriptive statistics for the interest variables are presented in Table 2.

**Table 2.** Summary statistics for the numerical variables.

|  | DEAL VALUE | STAKE | P/L | EBIT |
|---|---|---|---|---|
| Mean | 5006.11 | 13.34002 | 185,540.15 | 1,238,961.5 |
| Maximum | 958,528.63 | 1715.653 | −191,478.74 | −1,153,817.89 |
| Minimum | 2.38 | −2.590000 | 1,073,899.29 | 4,875,992.59 |
| Std. Dev. | 37,959.88 | 86.07237 | 275,023.11 | 1,600,562.54 |
| Skewness | 22.7 | 16.36611 | 1.59 | 1.23 |
| Kurtosis | 564.204 | 287.1457 | 1.89 | 0.193 |

Source: Own processing using SPSS 25.0.

The distributions for the two dependent variables, deal value and stake, were highly asymmetrical, with extremely high deviations from the mean. In order to smooth the series, we used the logarithmic values for these variables when estimating the regression models.

The distribution of the logarithmic values of stake (see Figure 1) was bimodal, and it can be observed that the small values had a clearly separated distribution from the one for the high values. This finding led us to estimate a conditional quantile regression model for stake, in order to identify a more accurate relationship with the regressors.

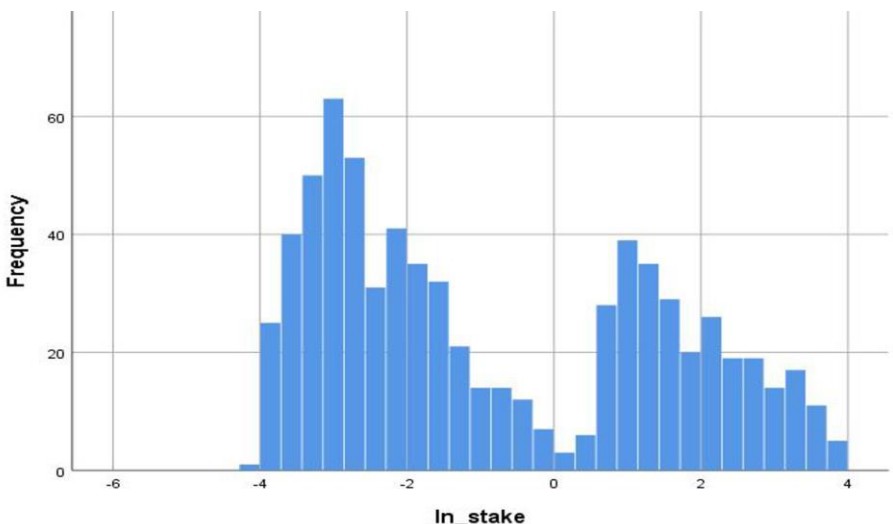

**Figure 1.** Distribution of companies by ln(stake). Source: Own processing using SPSS 25.0.

The descriptive statistics for the profile of the acquirer, as found in the sample, are presented in Table 3.

**Table 3.** Descriptive statistics for the acquirers' profile.

|  |  | Transactions with Listed Acquirer | Transactions with Unlisted Acquirer | Total |
|---|---|---|---|---|
| Year of the event | 2010 | 1 | 4 | 5 |
|  | 2011 | 0 | 4 | 4 |
|  | 2012 | 1 | 23 | 24 |
|  | 2013 | 3 | 48 | 51 |
|  | 2014 | 4 | 56 | 60 |
|  | 2015 | 8 | 50 | 58 |
|  | 2016 | 7 | 92 | 99 |
|  | 2017 | 10 | 191 | 201 |
|  | 2018 | 11 | 197 | 208 |
| Total |  | 45 | 665 | 710 |

Source: Own processing using SPSS 25.0.

Regarding other variables used in the study, their distribution and the descriptive statistics will add to the understanding of Romanian minority acquisitions. Thus, referring to the status of the acquirer, 45 companies were listed (representing 6.34% of the sample) and 665 companies were unlisted (out of which 208 were undisclosed acquirers, representing 29.30% of the total sample). Considering the year, the data describe the evolution of the Romanian market regarding minority acquisitions and is consistent with the increasing international trend. Thus, with few transactions after the financial crisis of 2008–2009, the last three years of the sample showed an exponential increase in the activity of investors in minority acquisitions. Regarding accounting practice, listed acquirers apply IFRS (according to Romanian regulations), and the unlisted ones apply local GAAP (457 investors, because we exclude the persons—208 acquirers).

In order to verify the research hypotheses, both models were estimated: an OLS regression to assess the influence of the financial performance of the target company, reflected in profit/loss, over the deal value, and a quantile regression, in order to analyse the influence of the efficiency of operational activity of the target company on the purchased stake. The results are presented below.

The impact of financial performance on the deal value was estimated through an OLS regression. The estimated model for the whole sample was highly heterogeneous, with no reliable estimates for the coefficients. Hence, in order to estimate the impact of financial performance on the price the acquirer agrees to pay in transaction, we analysed the data (see Figure 2), and we observed that the small deal values were consistently correlated with results that were very close to zero.

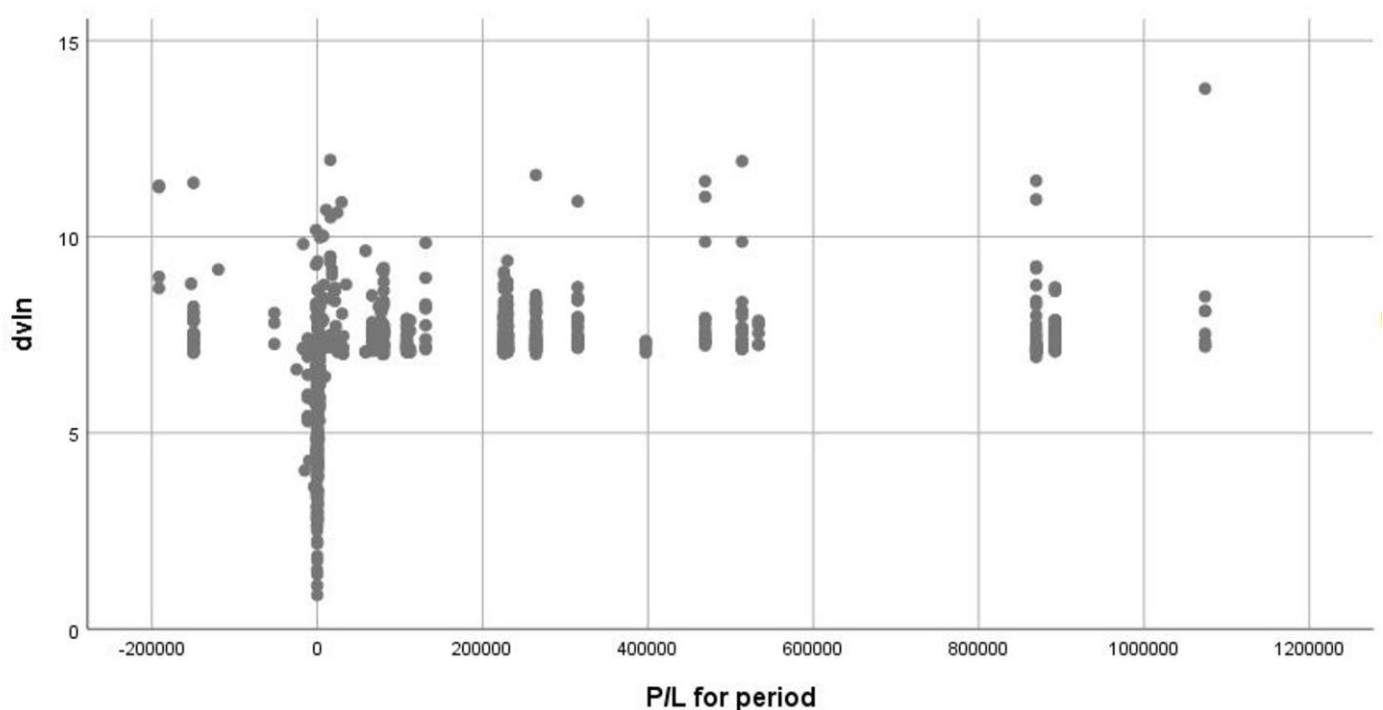

**Figure 2.** Correlation between deal value and profitability. Source: Own processing using SPSS 25.0.

In consequence, we divided the sample into two subsamples: one that contains the small transactions, with deal values as much as EUR 670.35 th ($\ln(670.35) = 6.5$), and one with the bigger transactions, with deal values over EUR 670.35 th. The results for the OLS regression, for the entire sample, are presented in Table 4.

**Table 4.** The estimated coefficients for the OLS regression—710 transactions.

| Model | Small Deal Values | | High Deal Values | |
|---|---|---|---|---|
| | Unstandardised Coefficients | Std. Error | Unstandardised Coefficients | Std. Error |
| Intercept | 3.622 *** | 0.692 | 9.290 *** | 0.773 |
| P/L | 0.000123 ** | 0.000 | $1.509 \times 10^{-6}$ *** | 0.000 |
| Listed_Acq | 0.247 | 0.734 | −1.921 ** | 0.768 |
| Unlisted_Acq | 0.324 | 0.700 | −1.812 ** | 0.747 |
| Ln_stake | 0.442 *** | 0.107 | 0.289 *** | 0.022 |
| Year_2010 | - | - | 1.842 *** | 0.392 |
| Year_2011 | - | - | −0.430 | 0.423 |
| Year_2012 | - | - | 0.539 ** | 0.250 |
| Year_2013 | 2.604 ** | 0.804 | −0.186 | 0.235 |
| Year_2014 | −0.031 | 0.378 | −0.063 | 0.227 |
| Year_2015 | −0.185 | 0.279 | −0.155 | 0.251 |
| Year_2016 | −0.477 * | 0.287 | 0.303 | 0.217 |
| Year_2017 | - | - | 0.427 ** | 0.210 |
| Year_2018 | 0.063 | 0.274 | 0.474 ** | 0.209 |

*** Significance at the 1% level; ** Significance at the 5% level; * Significance at the 10% level. Source: Own processing using SPSS 25.0.

The results show that the financial profitability had a significant positive effect in explaining the variation of both the small and high deal values, with an increase in the profit/loss of the target company significantly increasing the deal value. The difference was in the magnitude of the response; the effect of the reported income was substantially higher for the small transactions than for the ones in which the acquirers invested large amounts. These results confirm the behaviour of small investors, who purchase shares in profitable companies with the sole purpose of gaining financial revenues, in the form of dividends.

Given the scarce literature on M&A-related comparison of the three major economy sectors and the fact that the noise in one sector can be reduced at company level data (Shah and Shin 2007), we considered it necessary to eliminate from our analysis the primary sector and the strategic national companies (electricity, gas, and other similar companies, considered blue-chips for BSE, being included in the Bucharest Exchange Trading index). We anticipated a difference of behaviour for the primary sector compared to the other two in terms of importance of productivity-related indicators, since, in this case, the acquirers search for strategic assets and concessions, not profit. Moreover, the particularities of the primary sector, with its strategic assets (Amighini et al. 2013; Alfaro 2003), compared to industry and services sectors, which are more profit-oriented, impose as necessary the analysis of influence of profit or loss of the target on the decisions made by the acquirer on paying a specific price, considering the three main sectors. The shares that are purchased in blue-chip companies have as a sole purpose the financial gain of the owners, so there cannot be a suspicion of takeover. As a result, a number of 308 transactions were considered.

The results for the OLS regression, for the selected sample of 308 transactions, are presented in Table 5.

**Table 5.** The estimated coefficients for the OLS regression—308 transactions.

| Model | Small Deal Values | | High Deal Values | |
|---|---|---|---|---|
| | Unstandardised Coefficients | Std. Error | Unstandardised Coefficients | Std. Error |
| Intercept | 3.609 *** | 0.700 | 9.790 *** | 0.977 |
| P/L | 0.000125 ** | 0.000 | $2.406 \times 10^{-6}$ | 0.000 |
| Listed_Acq | 0.242 | 0.743 | −2.361 *** | 0.890 |
| Unlisted_Acq | 0.304 | 0.708 | −2.181 ** | 0.863 |
| Ln_stake | 0.449 *** | 0.109 | 0.173 *** | 0.062 |
| Year_2010 | - | - | 2.048 *** | 0.676 |
| Year_2011 | - | - | −0.049 | 0.664 |
| Year_2012 | - | - | 1.032 | 0.750 |
| Year_2013 | 2.652 *** | 0.816 | −0.161 | 0.554 |
| Year_2014 | −0.012 | 0.385 | 0.001 | 0.470 |
| Year_2015 | −0.167 | 0.285 | −0.117 | 0.469 |
| Year_2016 | −0.461 | 0.293 | −0.589 | 0.482 |
| Year_2017 | - | - | −0.244 | 0.449 |
| Year_2018 | 0.049 | 0.284 | 0.244 | 0.454 |

*** Significance at the 1% level; ** Significance at the 5% level. Source: Own processing using SPSS 25.0.

Thus, when excluding the primary sector and the companies that are preferred for transactions (blue-chip companies from utilities and the financial sector), for small stakes, the behaviour was similar, because it reflected the attitude of the small investors, who want to invest in order to obtain dividends because the company they invest in reports a high profit. When investing high amounts, the influence is still positive, but not significant, which means there is a takeover assumption.

For investigating the second dimension of the acquirer's behaviour, we considered two categories of stakes: small (0.001–1%) and large stakes (1.001–50%), for two reasons: on a side, we have different behaviours, in terms of motivation and decision autonomy, for small investors with insignificant shares in the target companies, and for strategic investors, which purchase important shares; on the other side, there are some restrictions regarding the maximum stake (of 1%) bought by a single investor in financial investment companies. The efficiency of the operational activity, reflected in EBIT, had a negative and significant influence on the small purchased stakes. When purchasing larger stakes, the efficiency of operational activity had a positive, but non-significant, influence on the purchased stake. This relationship is typical for the Romanian market for corporate control, characterised by many takeovers involving target companies that are undervalued and acquirers that purchase companies for their assets or their market share, results consistent with other studies (Ciobanu 2015; Pop 2006). Thus, the investors are interested in aspects other than operational profit.

The results from estimating both the OLS regression and the quantile regression models for the sample of transactions with stake percentages below 1% are presented in Table 6.

Figure 3 further shows the sensitivity of stake to changes in the regressors for transactions with stake percentage below 1%. For EBIT, there were differences in the response of the stake at extreme quantiles, especially at the high quantiles, when the response was positive. There were notable differences between the two estimates: for the lower quartiles, the OLS overestimated the response of the dependent variable, and for the high quantiles, the response was strongly underestimated by OLS. Therefore, the QR model led to more accurate estimates of the marginal effect of the regressors on stake, mainly for the high

quantiles, associated with acquirers that pursue the control of the target company. Also, for the lower and for the higher quantiles, the QR estimates differed significantly from the OLS estimate, with the latter being found outside the limits of the confidence intervals of the QR estimates.

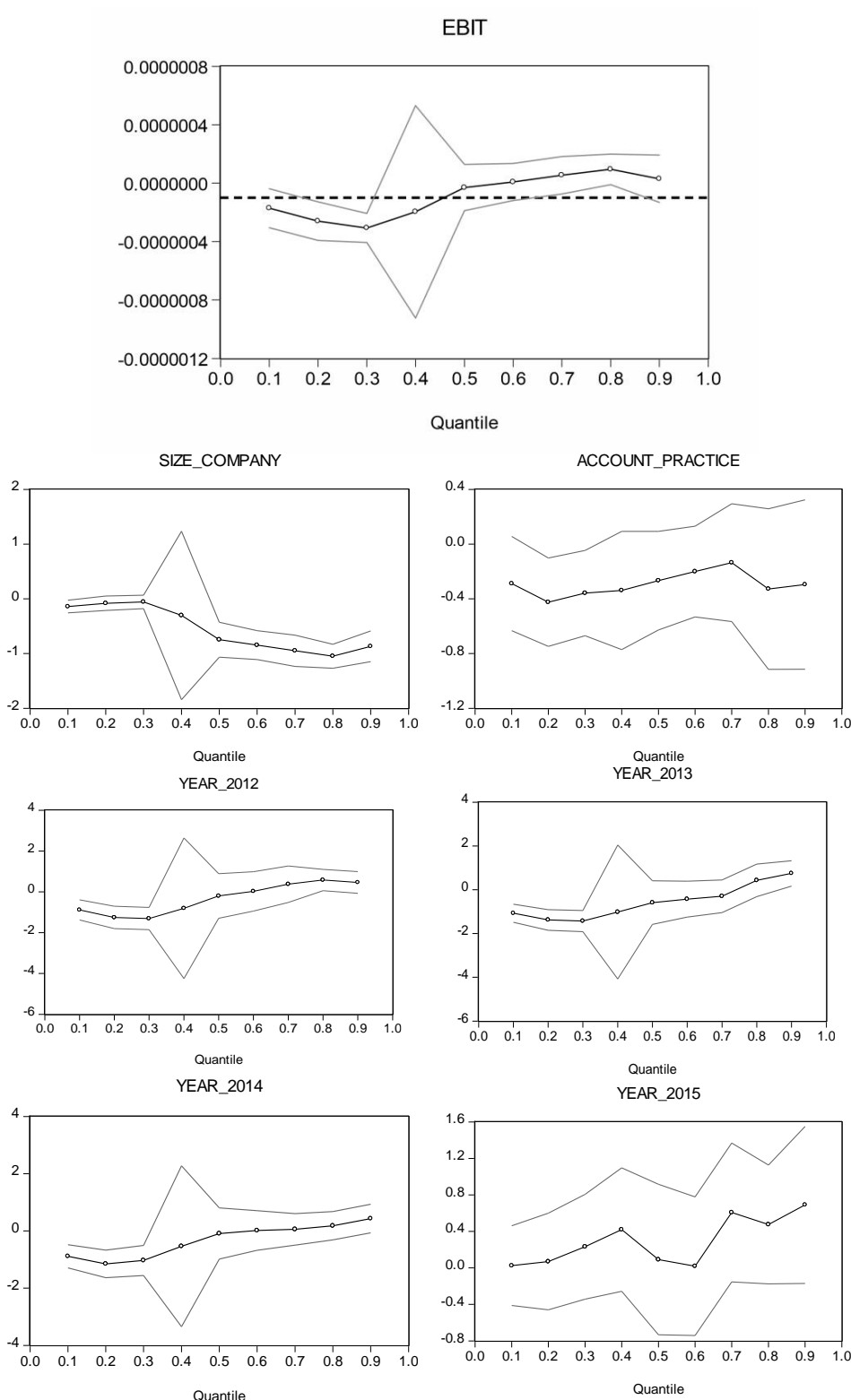

**Figure 3.** *Cont.*

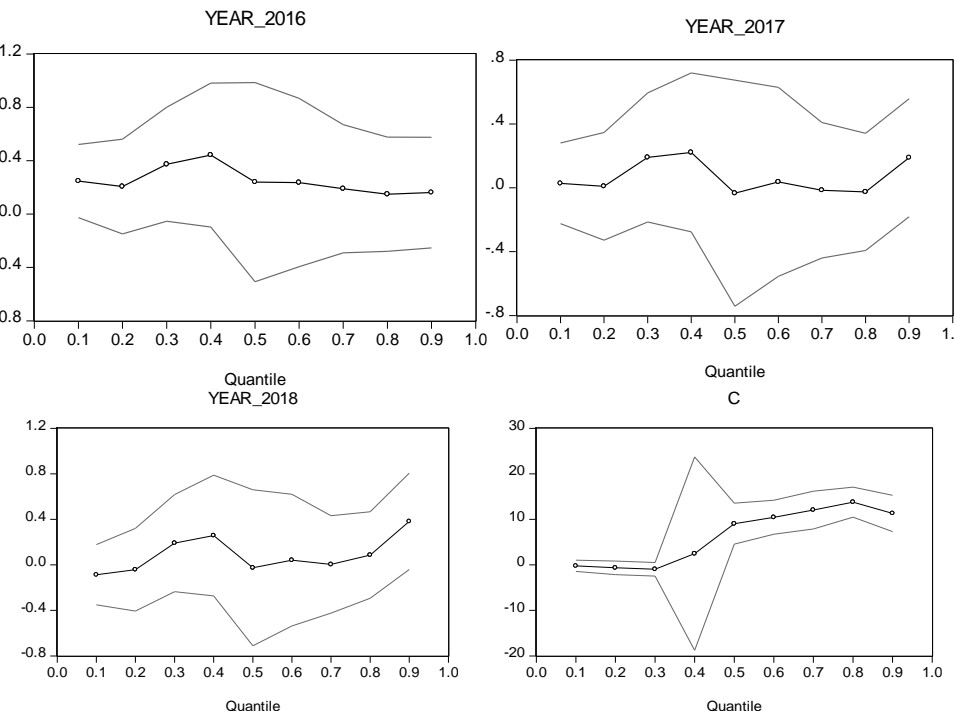

**Figure 3.** Impacts of changes in the explanatory factors on stake, across quantiles, for transactions with stake percentage below 1% (note: the middle line depicts the QR coefficient estimates, framed by the 95% confidence interval bounds, in grey; the dotted horizontal line represents the OLS estimate—presented only for the interest variable). Source: Own processing.

The results from estimating both the OLS regression and the quantile regression models for the sample of transactions representing stake purchases above 1% are presented in Table 7.

The OLS estimate of the EBIT's effect on stake was positive and not statistically significant at the 5% level, and it was the same for the QR estimates of this effect. An increase in the efficiency of operating activity would not generate a significant response from the value of a purchased stake, so that we can say that the transactions for a stake above 1% are substantiated on other motives than the efficiency of the operational activity (strategic assets, devalued companies, market share, etc.).

The results are also presented in Figure 4, showing, for the interest variable, that although there are some differences between the QR estimates and the OLS estimate, none of the differences were statistically significant, with the OLS estimate being covered by the confidence intervals bounds for the QR estimates.

**Table 6.** Stake's response to changes in the influence factors for transactions with stake percentages below 1%.

| | OLS | QR | | | | | | | | |
|---|---|---|---|---|---|---|---|---|---|---|
| | | 0.1 | 0.2 | 0.3 | 0.4 | 0.5 | 0.6 | 0.7 | 0.8 | 0.9 |
| Intercept | 5.225099 | −0.20636 | −0.63871 | −0.96330 | 2.477620 | 9.047370 | 10.45988 | 12.03112 | 13.77523 | 11.30477 |
| EBIT | $-1.07 \times 10^{-7}$ * | $-1.7 \times 10^{-7}$ * | $-2.6 \times 10^{-7}$ * | $-3.1 \times 10^{-7}$ * | $-1.9 \times 10^{-7}$ | $-3.03 \times 10^{-8}$ | $7.4 \times 10^{-9}$ | $5.3 \times 10^{-8}$ | $9.4 \times 10^{-8}$ * | $2.9 \times 10^{-8}$ |
| Size_company | −0.501442 | −0141668 | −0.081014 | −0.057310 | −0.303645 | −0.746653 | −0.845515 | −0.948767 | −1.049309 | −0.868217 |
| Account_practice | 0.028406 | −0.288334 | −0.424881 | −0.358097 | −0.338878 | −0.266959 | −0.200261 | −0.135252 | −0.328732 | −0.295174 |
| Year_2010 | - | - | - | - | - | - | - | - | - | - |
| Year_2011 | - | - | - | - | - | - | - | - | - | - |
| Year_2012 | −0.249502 | −0.8834 | −1.25333 | −1.30908 | −0.80464 | −0.20223 | 0.022501 | 0.374137 | 0.581815 | 0.462491 |
| Year_2013 | −0540109 | −1.07349 | −1.37991 | −1.43064 | −1.01624 | −0.58683 | −0.42775 | −0.29673 | 0.43058 | 0.746311 |
| Year_2014 | −0.164137 | −0.88808 | −1.15523 | −1.03242 | −0.53587 | −0.09058 | 0.015688 | 0.054518 | 0.181591 | 0.432259 |
| Year_2015 | 0.366284 | 0.0253 | 0.069599 | 0.231708 | 0.420049 | 0.091638 | 0.018256 | 0.607393 | 0.475788 | 0.690391 |
| Year_2016 | 0.352046 | 0.247019 | 0.20624 | 0.373213 | 0.441607 | 0.239168 | 0.236515 | 0.18986 | 0.148241 | 0.161041 |
| Year_2017 | 0.096108 | 0.028036 | 0.00917 | 0.190375 | 0.221712 | −0.0338 | 0.037378 | −0.01522 | −0.02571 | 0.18845 |
| Year_2018 | 0.034406 | −0.0865 | −0.04278 | 0.190784 | 0.257842 | −0.02564 | 0.04163 | 0.004005 | 0.086414 | 0.382249 |

Note: * Significance at the 5% level. Source: Own processing using SPSS 25.0.

**Table 7.** Stake's response to changes in the influence factors for transactions with a stake percentage above 1%.

| | OLS | QR | | | | | | | | |
|---|---|---|---|---|---|---|---|---|---|---|
| | | 0.1 | 0.2 | 0.3 | 0.4 | 0.5 | 0.6 | 0.7 | 0.8 | 0.9 |
| Intercept | 3.3384 | 1.320581 | 1.009045 | 3.007506 | 3.024673 | 3.350307 | 3.854606 | 4.367692 | 5.497101 | 4.969254 |
| EBIT | $6.2 \times 10^{-8}$ | $1.52 \times 10^{-7}$ | $2.14 \times 10^{-8}$ | $4.68 \times 10^{-8}$ | $9.31 \times 10^{-8}$ | $4.66 \times 10^{-8}$ | $2.32 \times 10^{-8}$ | $4.67 \times 10^{-8}$ | $6.68 \times 10^{-8}$ | $1.44 \times 10^{-7}$ |
| Size_company | −0.1205 | −0.10587 | −0.0657 | −0.10997 | −0.12472 | −0.1363 | −0.13901 | −0.19888 | −0.16701 | −0.10048 |
| Account_practice | −0.0662 | 0.055401 | −0.10464 | −0.08063 | 0.085151 | −0.08493 | −0.13736 | 0.061971 | −0.23174 | −0.39969 |
| Year_2010 | −0.0418 | 0.581727 | 0.796788 | −0.75978 | −0.78773 | −0.38476 | −0.80561 | 0.976174 | −0.30041 | −0.06911 |
| Year_2011 | −0.3282 | 0.64899 | 0.577351 | −0.67095 | −0.51662 | −0.70642 | −0.06762 | −0.01126 | −0.90612 | −1.04004 |
| Year_2012 | 0.2986 | 1.049962 | 0.800839 | 0.06381 | 0.12086 | 0.144175 | 0.752512 | 0.83033 | −0.27162 | −0.5084 |
| Year_2013 | −0.34838 | 0.348793 | 0.602882 | −0.72592 | −0.61857 | −0.53549 | −0.51848 | −0.4801 | −0.50785 | −0.14015 |
| Year_2014 | −0.2444 | 0.60826 | 0.818344 | −0.59777 | −0.31449 | −0.23791 | −0.56107 | 0.060115 | −0.97407 | −0.6249 |
| Year_2015 | −0.1163 | 0.492919 | 0.710173 | −0.5834 | −0.41023 | −0.11471 | −0.02248 | 0.210077 | −0.74759 | −0.60662 |
| Year_2016 | −0.1617 | 0.494301 | 0.781366 | −0.44034 | −0.2215 | −0.13892 | −0.41681 | −0.02086 | −0.7997 | −0.65717 |
| Year_2017 | −0.5185 | 0.466846 | 0.606761 | −0.84566 | −0.59783 | −0.50744 | −0.64983 | −0.50697 | −1.52852 | −1.03712 |
| Year_2018 | −0.1702 | 0.507339 | 0.795558 | −0.52607 | −0.19638 | −0.07138 | −0.31797 | −0.02868 | −0.96438 | −0.58571 |

Note: Source: Own processing using SPSS 25.0.

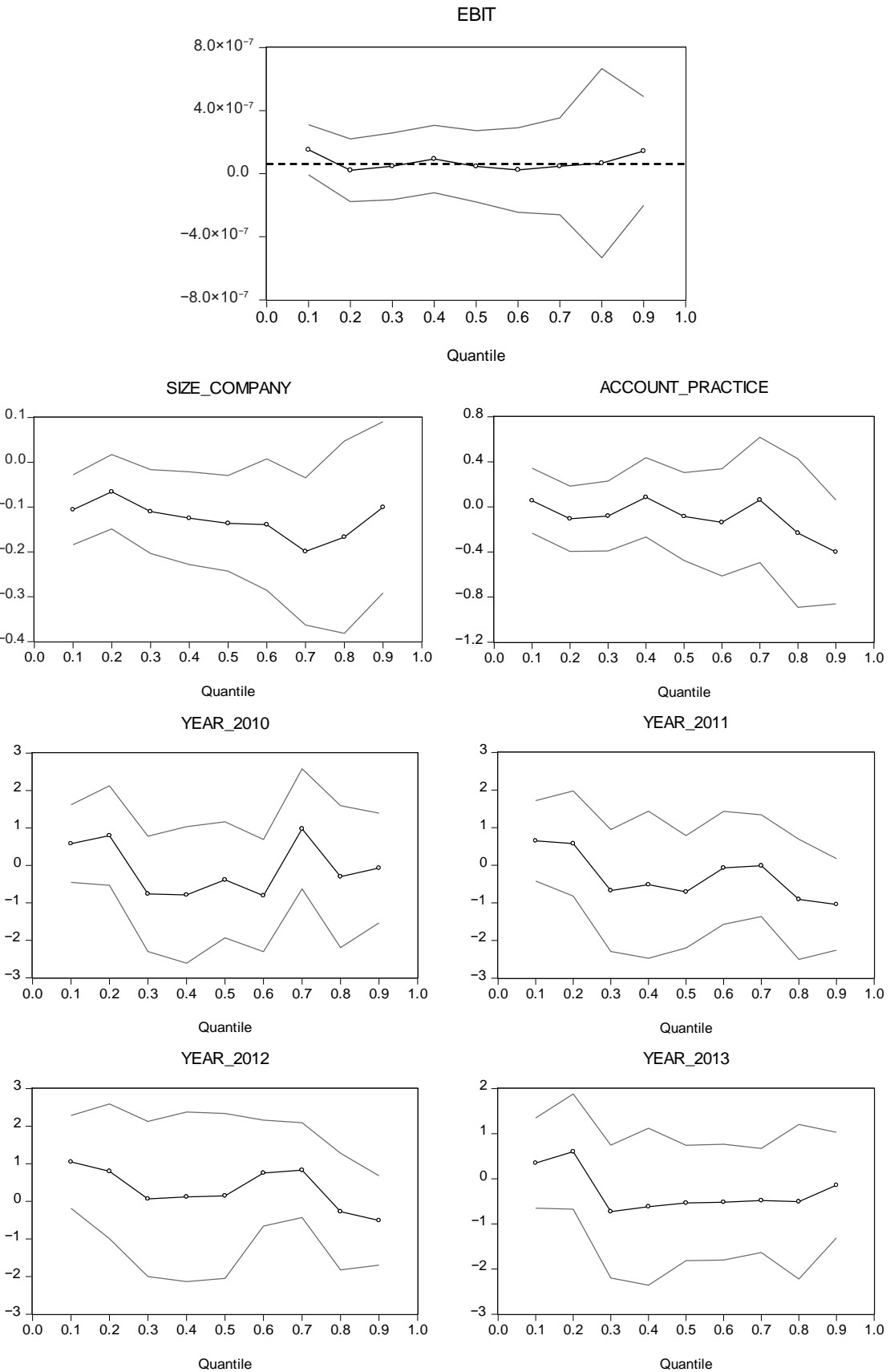

**Figure 4.** *Cont.*

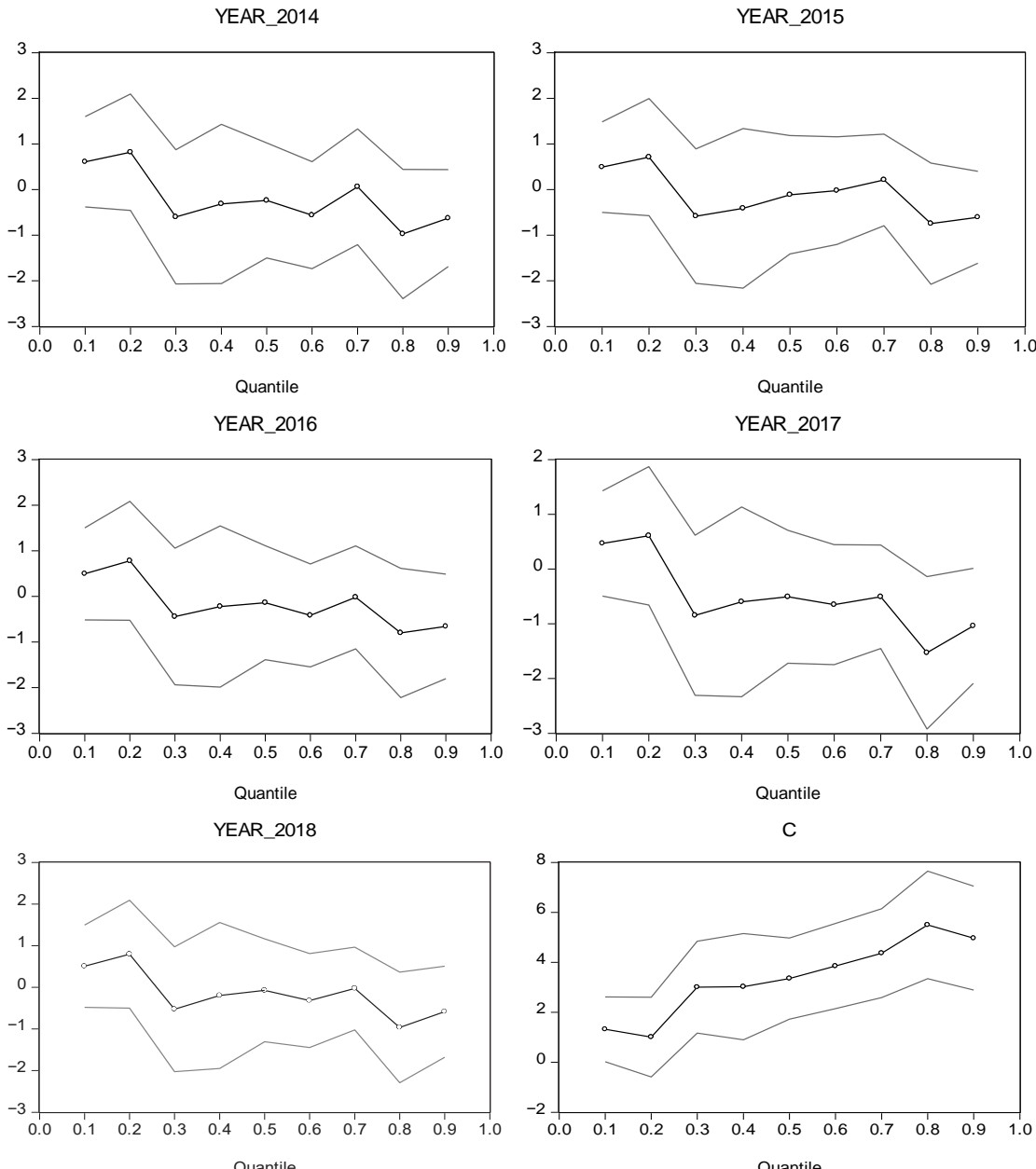

**Figure 4.** Impacts of changes in the explanatory factors on stake, across quantiles, for transactions with stake percentage above 1% (note: the middle line depicts the QR coefficient estimates, framed by the 95% confidence interval bounds, in grey; the dotted horizontal line represents the OLS estimate—presented only for the interest variable). Source: Own processing.

When analysing the behaviour of the investors on BSE—how much to spend and how much to buy from a target company—considering only the minority acquisitions, under the influence of performance factors related to the latter, the results describe two types of acquirers as a response to the two working hypotheses. On one side, we have the investments made with the main purpose of generating dividends or to be sold when their market price rises. Given their potential to improve the acquirers' cash-flow, we analysed whether the financial performance of the target company significantly influenced the investment decision of the acquirer in terms of the amount spent. The results showed that, when paying low deal values, the investors are usually unlisted and with high interest for the performance of the target company. Their profile, in this case, is that they are interested in raising their cash-flow and recording capital gains, results that are consistent with the

ones of Nguyen et al. (2022) and Agyei-Boapeah et al. (2019), who also concluded that the lower the investment, the higher the interest in increasing financial gains or reducing financial constraints of the acquirers, under the influence of the target's performance. When the deal value is increasing, the financial performance stops being a significant influence factor for the investors' decision to spend a high amount of cash, which, in our opinion, leads to the second objective for participating in minority M&As; the decision to keep the shares; and the decision to accumulate a stock of shares which allow for the influence of the target company or, in time, its takeover (Pinelli et al. 2020). The first hypothesis is validated. The influence is also tasted on a sample of 308 transactions, which resulted after excluding from the initial sample the investments in the primary sector and in blue-chip companies. In this case, the profit and loss stops being a significant influence factor which, in our opinion, underlines once more the intention for financial gains in the case of small deal purchases. Also, the assumption of accumulating shares and/or takeover appears once more when high amounts are paid, representing deal values. The results are consistent with other findings in the literature (Eckbo 2009; Alexandridis et al. 2013; Dicu et al. 2019).

Considering the second hypothesis and the chain of thoughts, we analysed whether the decision to acquire specific stakes in the target companies listed on the BSE is made with the intention to keep the stock of shares and/or to take over the target company, under the influence of the efficiency of the operational activity reported by the acquired company. The results show that, when the investors acquire more than 1% of the target company, the operational profitability is not a factor with significant influence. This result leads us to conclude that, in the case of purchasing higher stakes in the target companies, the motives and the behaviour of the acquirer are more oriented towards accumulating stock of shares in order to access the assets of the target, to influence its activity, etc. The second hypothesis was not validated. Our findings are consistent with other studies found in the literature (Liao 2014; Almeida and Gomes Novaes 2020; Kang and Kim 2008).

## 5. Conclusions

The empirical results of our study assess, on one side, the conditions in which an acquirer's decision to place its funds in a target company's shares is influenced by the latter's financial performance, and, on the other side, to which extent the assumption to take over a target company is influenced by the efficiency of the operational activity of the acquired company, the acquirer intending to invest in a company that results in efficiency gains, or synergy success.

Our empirical findings suggest that when a buyer is interested in paying low amounts of deal value, they are more focused on financial gains than when they are willing to pay high amounts of its funds. Excluding from our sample the blue-chip companies and the target companies from the primary sector, the assumption to takeover appeared once more when paying high amounts of deal value.

Further in-depth studies show that, in the case of purchasing a stake below 0.5%, the investment decision is not necessarily based on the own assessment of financial gains, but mainly on financial counselling. Purchasing a stake between 0.5 and 1% shows a positive and significant relationship between the amount of shares acquired and the interest of the investor in the efficiency of operational activity, which makes us assert that acquirors are paying, in the case of Romanian listed companies, a closer look at the financial information. Our study suggests that this occurs for gaining access to sensible data or obtaining more in-depth financial information on the target company, which may lead to further strategic acquisitions.

Further, the analysis shows that purchased stakes above 1% may underline different motives for acquisitions, such as market share, taking over strategic assets, vertical/horizontal integration, and competition issues.

One of the limits of the study arose as a result of the low quantity of data available for the transactions undertaken on the Bucharest Stock Exchange, especially in the case of financial data for unlisted acquirers, because the behaviour of the investors might be

influenced, especially in the case of companies, by their financial status. Also, limited or poor disclosure of the companies, especially small and medium ones, and of the institutions, which is one of the characteristics of an emerging economy, may have influenced the reported number of transactions in the early 2010s.

Finally, several questions arose. Our results show two types of acquirer behaviour under the influence of the target's financial performance: one type of investor oriented towards financial gains and increased cash-flow, and the other type, which accumulates stock of shares with different intentions. Therefore, future research could further examine the behaviour of acquirers in their minority acquisition activities, considering the second situation and the financial information on the target companies. As such, two questions raised are whether the financial market indicators or the capital structure of the target affect the bidders' equity ownership choices and amounts to be invested.

**Author Contributions:** Conceptualization, R.-M.D. and L.-G.M.; methodology, E.-D.V. and G.-M.A.; software, E.-D.V.; validation, E.-D.V. and G.-M.A.; formal analysis, L.-G.M.; investigation, R.-M.D.; resources, G.-M.A.; data curation, E.-D.V.; writing—original draft preparation, R.-M.D., L.-G.M. and G.-M.A.; writing—review and editing, R.-M.D. and G.-M.A.; visualization, L.-G.M.; supervision, L.-G.M.; project administration, L.-G.M.; funding acquisition, L.-G.M. and R.-M.D. All authors have read and agreed to the published version of the manuscript.

**Funding:** This study was supported by the Erasmus+ programme of the European Union under the Jean Monnet Chair, 'Doing Resilient Business on the European Market' (DO RE BIZ) [project number ERASMUS-JMO-2022-HEI-TCH-RSCH-101085838].

**Informed Consent Statement:** Not applicable.

**Data Availability Statement:** The data presented in this study are available on request from the corresponding author. The data are not publicly available due to Zephyr and Orbis databases policies.

**Conflicts of Interest:** The authors declare no conflict of interest.

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
