# Peer review of "Determinant Factors of M&As in Emerging Economies: The Impact of Financial Performance in Romanian Minority Acquisitions"

_economies, doi:10.3390/economies11100241_

Round 1

Reviewer 1 Report

Review on

 Determinant factors of M&As in emerging economies: The acquires’ behaviour in Romanian minority acquisitions

1.     Abstract

The abstract should be expended in order to include also the methodological issues.

2.     Methodology

The period refer to 2010-2018 period. The authors explained ehe ending period 2018 based on the Brexit causes. Maybe the authors should developed better these explanations. I do not see strong relationship between Brexit and the acquisition of  minority shares in Romanian listed target companies?

I suggest to put a table with the description of variables including the name of variables, way of calculation, description and sources. It is difficult to follow how the description of variables are correlated with the symbols. What is P/L? It suppose to be the dependent variables but I didn’t see it at the section of dependent variable.

Then, I have some concern regarding the way in which the independent variable is approached.

The financial performance include both net results and EBIT. I do not understand why the authors split financial performance as being net results and efficiency of activity being EBIT. Actually, performance is a wide concept that include both indicators. However,

both indicators are expressed in absolute values, which may create bias in the results related to the size of the companies. As absolute values they may not offer comparable analysis. A ratio would have been very welcome (Net results/Total assets and EBIT/total assets for instance) in order to avoid any bias of size of the companies.. This ways these ratios are return ratios. Instead total assets the turnover should be used. In this case, they became profit margin ratios, very useful for determining the profitability of a company.

I suggest to use the term financial performance as being the independent variable. Then, this variable should be measured  systematically by Net Result and by EBIT.

Then, I do not understand why if the independent variable is financial performance, why the title of the paper is not related with this. Term of behavior suggest anything than a relationship between mergers& acquisition and the relationship with the financial performance . Maybe the title should be changed such as: The impact of dinancial performance on M&A . This is more suggested as keywords to express better the content of the paper.

2 Results

The presentation of outputs from the statistic program should be done not with copy paste, but they have to be rewritten in separate tables/ I suggest that summary statistics to be conducted for all the variables included in the study.

I do not understand the Figure 2. Correlation between deal value and profitability . Deal value shoul be on OY ax because it is dependent variable, while profitability shoul be on OX ax because it is explanatory variable, right?

3.     Discussion

The section of discussion is almost inexistent while the results should be aligned with the previous findings and better discussed.

 There is a lot of work in this paper but it requires a better organization of the methodology around the general idea of this research.

I am not a native En speaker but I found a good written En, from my experience.. No typos were identified. 

Reviewer 2 Report

Thank you for the opportunity to review the manuscript submitted to Economies.

The paper must be improved in line with the comments below:

1. The bibliography is older, there should be at least 14 new sources from the last 3 years (minimum 20% of the bibliography).

2. The author needs to mention the limitations of the study.

3. All figures and tables must include the source.

I hope you find the above comments useful, and I wish you the best of luck with developing the paper further.

Reviewer 3 Report

Dear Authors,

I read your paper and have some issues with it:

1. I fail to understand your definition of M&A. On one side you discuss about investments made in minority stakes for the purpose of earning dividends and capital gains and on the other side you analyze possible attempts at takeovers. How did you decide the former? Any individual investing even in one share of a listed Romanian company is considered to be having a minority stake?

2. While there are less than 300 listed companies in the Bucharest Stock Exchange, you claim to have collected data on 308 samples. How many M&A did really take place in Romenia? Out of the data of 308 samples, how many were investors holding a few shares as an investment and how many were speculators trying to make a fast buck with the market volatility?

3. While your analysis looks statistically sound, I think the methodology used to identify the samples is totally unacceptable.

4. It would be better the analyze only the real M&As that really took place in the country and their effect on the GDP or on the market capitalization, rather than considering every trade in the stock exchange as a minority stake.

The English looks good.

Round 2

Reviewer 3 Report

Dear Authors,

I think your article is good enough to be published now.

A final overall review of the language should be done.